# Acute Blood Pressure Changes Following Resistance Exercise in Adults with Hypertension

**DOI:** 10.3390/jfmk10030349

**Published:** 2025-09-12

**Authors:** Luis A. Benavides-Roca, Germán Parra, Antonio R. Zamunér

**Affiliations:** 1School of Sports Sciences and Physical Activity, Faculty of Health, Universidad Santo Tomas, Avenida Carlos Schorr 255, Talca 3473620, Chile; benavides.roca@gmail.com; 2Faculty of Education, Universidad Autónoma de Chile, 5 Poniente 1670, Talca 3467987, Chile; 3Department of Physiology, Universidad de Valencia, Avenida de Blasco Ibáñez, 15, El Pla del Real, 46010 Valencia, Spain; german.parra@uv.es; 4Laboratory of Clinical Research in Kinesiology, Department of Kinesiology, Universidad Católica del Maule, Avenida San Miguel 3605, Talca 3480112, Chile

**Keywords:** resistance exercise, cardiovascular system, blood pressure, hypertension

## Abstract

**Objective**: This study aimed to determine the acute effects of specific exercises, commonly used in resistance training programs, performed at two intensities (low and high) with equal volume, on blood pressure in individuals with pharmacologically controlled hypertension. **Methods**: A crossover clinical trial was conducted with 26 participants with hypertension, each completing two resistance training sessions, one low-intensity and one high-intensity, including squats, rows, deadlifts, and bench presses. The exercise intensity for the first session was randomly assigned, and blood pressure was measured using an automated device before and immediately after each exercise. Comparisons between baseline and post-exercise values were analyzed, with a significance level set at 5%. **Results**: The results showed that systolic blood pressure (SBP) increased during squats, rows, and deadlifts compared to baseline, with the deadlift producing the highest values. For diastolic blood pressure (DBP), no significant differences were observed from baseline, but the squat and deadlift elicited higher values than the rows and bench press. Additionally, high-intensity training resulted in lower DBP values compared to the low-intensity protocol. **Conclusion**: Resistance exercise increases SBP regardless of intensity, with the squat and deadlift producing the most significant changes, while high-intensity training leads to lower DBP values than low-intensity protocols in the studied participants.

## 1. Introduction

Exercise is considered as one of the main non-pharmacological therapies for patients with hypertension [1]. Therefore, different modalities of exercise have been studied, including resistance training. Systematic reviews [2,3] have shown that resistance exercise contributes to the reduction in blood pressure in stages of prehypertension and hypertension. Furthermore, the post-exercise hypotensive effect that lasts over time has been observed [4,5], especially when exercise is of moderate intensity [3]. Mechanisms such as nitric oxide release and decreased adrenergic discharge, which result in lower peripheral vascular resistance [6], have been proposed as contributing factors. However, studies evaluating the acute effects of specific resistance exercises at different intensities are incipient [7,8]. There is evidence that resistance exercise produces an increase in blood pressure immediately after performance [7,8], showing an increase of approximately 20 mmHg in systolic (SBP) and 10 mmHg in diastolic blood pressure (DBP) with exercises such as inverted rows, squats, push-ups, and sit-ups. Similarly, in hypertensive subjects, resistance exercise at 80% intensity has been shown to increase SBP by approximately 6 mmHg, which is higher than the increase observed at 40% intensity [9]. It is worth noting that several factors have been shown to influence the blood pressure response to exercise. Ishikawa et al. [10] reported that age, but not sex, affects this response. Similarly, Forjaz et al. [11] observed that, in individuals with hypertension, blood pressure response is influenced by age, body mass index, baseline blood pressure, and cardiorespiratory fitness.

Enhancing knowledge of the immediate effects of specific exercises on blood pressure, while taking into account possible confounding factors, may help patients and therapists identify exercises that elicit greater increases in blood pressure in this population. Therefore, the hypothesis of the study was that high-intensity resistance training involving larger muscle groups would produce greater elevations in blood pressure than low-intensity exercises involving smaller muscle groups, regardless of sex, age, body mass index (BMI), and medication use. Accordingly, this study aimed to determine the acute effects of specific exercises commonly used in resistance training programs (i.e., squats, rows, deadlifts, and bench press), performed at two intensities (low and high) with equal training volume, on blood pressure in individuals with pharmacologically controlled hypertension.

## 2. Materials and Methods

### 2.1. Study Design

The present study is a secondary analysis of a randomized, single-blind, crossover clinical trial, registered at ClinicalTrials.gov (NCT06370546).

### 2.2. Participants

Sample size was calculated a priori using G*Power (version 3.1) software based on the study by Orsano et al. [12], considering SBP and DBP as main outcomes. Twenty-four participants were suggested, considering a medium effect size (f = 0.25), with a significance level of 5% and a statistical power of 80%. Therefore, twenty-six participants (both male and female), aged between 30 and 60 years, with a mean body weight of 79.8 ± 13.1 kg and a height of 165.5 ± 7.6 cm, took part in the study Table 1. All participants had a confirmed medical diagnosis of hypertension established by a cardiologist, in accordance with the 2023 European Society of Hypertension (ESH) guidelines [13]. At the time of the study, all individuals were undergoing pharmacological treatment. In addition, participants must not have engaged in regular physical activity for at least three months prior to the start of the study.

**Table 1 jfmk-10-00349-t001:** Demographic characteristics of participants.

	Participants n = 26
Age (years)	47.3 ± 9.6
Male/female	10/16
BMI (kg/m^2^)	29.1 ± 0.1
Medication use (%)	
Angiotensin receptor blocker	65.4%
ACE inhibitors	15.4%
Calcium channel blockers	7.7%
Diuretics	7.7%
Beta-blocker	3.8%

BMI: body mass index; ACE: angiotensin-converting enzyme.

Participants were excluded if they had any neurological, musculoskeletal, or cardiovascular disease other than hypertension (e.g., atrial fibrillation, arrhythmias, coronary disorders, and pacemakers). In addition, participants were not included if they presented any contraindication for resistance exercise, were unable to perform the prescribed exercises, or reported to be drug abusers.

The study was performed in accordance with the Declaration of Helsinki, was approved by the local institutional ethics committee at the University Catholic of Maule (N°248- date 2022), and all participants signed an informed consent. Study recruitment is depicted in Figure 1 following the CONSORT flow diagram.

### 2.3. Procedure

All participants attended the Laboratory of Clinical Research in Kinesiology at the Catholic University of Maule on three occasions. In the first visit, inclusion and exclusion criteria were reviewed, and participants meeting the criteria were invited to take part in the study and signed an informed consent. A complete anamnesis was conducted to document past and current health conditions, physical activity habits, and medication use. Anthropometric characteristics were measured, including body mass and height, using a properly calibrated weight scale (Seca model 720, Hamburg, Germany). Then, the participants were acquainted with the experimental protocol and were requested to return on two other occasions to perform the resistance exercise sessions.

### 2.4. Randomization and Blinding

In the second visit, a randomization was performed to determine the intensity of the intervention (low-intensity or high-intensity resistance training). After a 1-week washout period, participants were first allocated to perform the low-intensity protocol and then returned to the laboratory to perform the high-intensity protocol, and vice versa.

### 2.5. Resistance Training

Two resistance velocity-based training protocols, with low- and high-intensity, were conducted using free weights. The low-intensity session comprised 6 sets of 12 repetitions performed at an intensity of 40% of one-repetition maximum (1RM, faster velocity of execution). In contrast, the high-intensity session consisted of 6 sets of 6 repetitions at 80% of 1RM (slower velocity of execution). The protocols were designed to equalize the total volume of the resistance training sessions [14]. The load required to reach the prescribed intensity was individually determined based on the velocity of execution, as proposed by Hernandez-Belmonte et al. [15] and Benavides-Ubric et al. [16]. Therefore, participants were instructed to perform each repetition at their maximal intended velocity, which was continuously monitored in real time using a previously validated linear transducer (ADR^®^, Toledo, Spain) [17].

Before the intervention, a 7–10 min warm-up was conducted, consisting of joint movements for the upper and lower body, aligned with the exercises included in the resistance training. During the warm-up, the execution speed for each exercise was measured as part of the approximation series. This evaluation allowed for the determination of the appropriate weights for each participant, tailored to the required session intensity. If the maximal velocity exceeded the pre-established velocity for a specific exercise, the workload was adjusted by increasing the weight until the predetermined velocity was reached.

The exercises were randomly assigned in an alternating sequence of lower- and upper-body movements, and this sequence was maintained throughout all subsequent sessions. Figure 2 shows the following: (1) Squats: consisting of a triple flexion of the lower limb (ankle, hip, and knee) with external resistance placed on the back using a barbell (execution velocity: low intensity = 1.15 ± 0.09 m/s; high intensity = 0.63 ± 0.05 m/s). (2) Reclined rows: participants were standing, maintaining the trunk flexion. Participants were instructed to bring the barbell close to the abdomen with elbow flexion and shoulder extension (execution velocity: low intensity = 1.42 ± 0.10 m/s; high intensity = 1.23 ± 0.07 m/s). (3) Deadlift: Participants lifted a bar from the ground while maintaining the core contraction and performing trunk, knee, and hip extensions (execution velocity: low intensity = 1.02 ± 0.09 m/s; high intensity = 0.57 ± 0.05 m/s). (4) Bench press: participants were instructed to stay in supine posture on a bench, with both feet on the ground, and to raise a barbell by flexing the shoulders and extending the elbows (execution velocity: low intensity = 1.14 ± 0.09 m/s; high intensity = 0.50 ± 0.06 m/s).

After each exercise, participants rested until they felt fully recovered, and hemodynamic variables returned to baseline values (approximately 10 min).

### 2.6. Main Outcomes

The main outcomes measured were blood pressure, which was recorded before the warm-up, after a prolonged rest of 10 min in a supine position, and after completing the series of each exercise.

The measurements included SBP and DBP, taken using a digital automated device (Omron HEM-907 sphygmomanometer, Healthcare, Tokyo, Japan).

### 2.7. Statistical Analysis

The SPSS Statistics version 22 (New York, NY, USA, IBM Corp.) program for Windows was utilized to perform the analyses. The Shapiro–Wilk test was applied to check the normality of the data. A two-way repeated measures analysis of variance (ANOVA) was performed to assess the interaction between intensity (high vs. low) and time (baseline vs. exercise). The significance level was set at 5% for all analyses. Cohen’s d was calculated to estimate the effect size of changes in SBP and DBP from baseline to immediately after each exercise, and was classified as negligible (<0.2), small (0.20–0.49), medium (0.50–0.79), or large (≥0.80) [18]. In addition, to explore potential confounding factors, linear regression analyses were conducted with SBP and DBP delta changes (post-exercise minus baseline) as the dependent variables, and sex, BMI, age, and medication use as independent variables.

## 3. Results

Table 2 presents the blood pressure values for each exercise and the intensity at which they were performed. The analysis showed no significant interaction between intensity and exercise for SBP (F_4,96_ = 1.35; *p* = 0.25) and DBP (F_4,96_ = 1.9; *p* = 0.12). However, a significant main effect of exercise was observed for all outcomes (SBP and DBP [*p* < 0.001]).

Regardless of the intensity, the data for SBP indicate that the baseline values were lower than those recorded during the squat (*p* < 0.001), rows (*p* < 0.001), and deadlift (*p* < 0.001). Additionally, the bench press yielded lower values compared to the squat (*p* < 0.001), rows (*p* = 0.01), and deadlift (*p* < 0.001). Furthermore, the deadlift produced higher SBP than the rows (*p* = 0.01). DBP data indicate that, regardless of intensity, none of the exercises showed significant differences compared to baseline. However, the squat elicited higher values than the rows (*p* = 0.001) and the bench press (*p* < 0.001). Similarly, the deadlift exhibited higher pressure values compared to the rows (*p* = 0.001) and the bench press (*p* < 0.001). In addition, a significant effect of intensity was found for DBP (*p* = 0.003), indicating that high-intensity training resulted in lower values compared to the low-intensity protocol.

To account for a possible effect of sex, two- and three-way ANOVAs were performed to assess the interactions between intensity, exercise, and sex. No significant three-way interaction (exercise × intensity × sex) was found (SBP: F_4,96_ = 1.1, *p* = 0.37; DBP: F_4,96_ = 1.2, *p* = 0.28), nor were there two-way interactions, i.e., exercise × sex (SBP: F_4,96_ = 1.0, *p* = 0.39; DBP: F_4,96_ = 0.5, *p* = 0.71) or intensity × sex (SBP: F = 0.23, *p* = 0.64; DBP: F = 0.18, *p* = 0.68).

Table 3 presents the delta changes in SBP and DBP from baseline to immediately after each exercise, along with the corresponding effect sizes. Increases in SBP with moderate to large effect sizes were observed after squat, rows, and deadlifts. In contrast, small effect sizes were found for DBP after high- and low-intensity exercises.

To assess the associations between SBP and DBP delta changes with potential confounding factors, linear regression analyses were conducted with SBP and DBP as the dependent variables, and sex, BMI, age, and medication use as independent variables. Full analyses are presented in the Appendix A. Significant associations were found between blood pressure changes after deadlift and bench press with medication use (Table 4). Using diuretics was linked to a higher increase in systolic BP after low-intensity deadlift (r^2^ = 0.17, B = 17.63, *p* = 0.04) and a higher increase in DBP after high-intensity bench press (r^2^ = 0.17, B = 9.54, *p* = 0.04). Use of ARBs was significantly associated with a decrease in DBP after low-intensity deadlift (r^2^ = 0.23, B = –9.78, *p* = 0.01) and high-intensity bench press (r^2^ = 0.19, B = –6.75, *p* = 0.02). Using ACE inhibitors was associated with a higher increase in DBP after low-intensity deadlift (r^2^ = 0.17, B = 10.91, *p* = 0.04) and high-intensity bench press (r^2^ = 0.16, B = 7.48, *p* = 0.05). Finally, the use of calcium channel blockers was associated with a higher increase in DBP after high-intensity bench press (r^2^ = 0.17, B = 9.54, *p* = 0.04). No significant associations were found between blood pressure changes and sex, BMI, and age.

## 4. Discussion

The purpose of this study was to examine the acute effects of specific resistance training exercises on blood pressure, performed at two intensities (low and high). The main findings were as follows: (1) squats and deadlifts elicited the highest SBP and DBP, regardless of intensity; (2) DBP was lower after high-intensity exercises compared with the low-intensity protocol; and (3) medication use appeared to influence blood pressure responses to specific exercises, particularly the deadlift and bench press.

These findings are in agreement with the results reported by Bond et al. [7], who evaluated subjects with prehypertension during handgrip exercises. The observed cardiovascular effects are consistent with the scientific statement from the American Heart Association [19], which highlights that during both dynamic and isometric exercise in healthy individuals, SBP tends to increase due to a rise in cardiac output, while DBP either remains stable or decreases slightly. This response is attributed to the interplay between cardiac output and peripheral vascular resistance, which collectively regulate the increment of blood pressure [20].

These results are important to characterize the blood pressure response immediately after exercises and intensities commonly used in clinical settings or in the context of cardiovascular rehabilitation, considering that the American College of Sports Medicine [21] recommends resistance exercise to be prescribed between 60% and 80% of 1RM while the European Society of Cardiology recommends low-to-moderate intensities (40% to 60% of 1RM) [22] for patients with hypertension. The hypertensive response to the resistance training observed in this study is in accordance with previous studies [23,24]. It is well established that physical exercise increases heart rate and stroke volume to meet the demands of the muscles. This is supported by Vale et al. [25], who observed a significant rise in SBP immediately after exercise with loads of 6 RM (high intensity) and 15 RM (low intensity). This SBP response was also observed in older women with hypertension, with SBP increasing by approximately 10 mmHg following exercises such as bench press, leg press, and lat pull-downs (3 sets of 10 repetitions) [23].

Based on these findings, the absence of significant differences between the high- and low-intensity groups after each exercise may be explained by the design of the strength training sessions, in which training volume was equalized despite differences in intensity. These findings highlight the importance of training volume as a crucial factor in strength training programming [26]. Effective volume management is essential for eliciting specific neuromuscular adaptations [27] and inducing blood pressure adjustments in hypertensive individuals [28]. Previous research has shown that similar neuromuscular adaptations can occur at different training volumes when the workload is equalized between groups [29]. These results are consistent with those observed in the present study, where training volume was equal across groups. Furthermore, longitudinal studies have demonstrated statistically significant reductions in blood pressure in sedentary patients using low training volumes (two sets per muscle group and exercise). For instance, a decrease in mean SBP from 152 mmHg to 122 mmHg and DBP from 83 mmHg to 73 mmHg was observed after 24 sessions over 12 weeks [28]. This suggests that training volume is a modifiable variable that can induce cardiovascular adaptations. Another possible factor that may have contributed to the absence of differences between high- and low-intensity exercises might be related to the velocity of movement execution. Previous studies have shown that high-velocity resistance exercise could recruit more muscle fibers, especially in those muscle groups with a large proportion of type II fibers (fast switch contraction), eliciting higher cardiovascular response, such as an increase in SBP or DBP [30,31]

For DBP, the response observed differs from that of SBP, as the results show no significant difference from baseline. This is consistent with the findings of Stöhr et al. [32], who also reported no significant changes in diastolic parameters (such as end-diastolic volume and early and late peak diastolic blood velocity) after a leg press session at both high (60% of 1RM) and low intensity (30% of 1RM). However, the literature presents mixed results. Marques et al. [33] observed an increase in DBP with both high (6.8% increase) and low (5.9% increase) training volume strength stimuli in older adults (78.9 ± 7.2 years old). Conversely, decreases in DBP (−18 mmHg) have been observed in individuals with hypertension and diabetes following strength training at 75% of 1RM [34]. Similarly, a reduction in DBP during the 15 min after self-loaded exercise (5.7 ± 1.5 mmHg) has been reported in physically active hypertensive individuals [8]. Therefore, it is essential to analyze DBP outcomes in relation to both the type of stimulus applied and the specific characteristics of the load, as these factors can influence key physiological mechanisms, including reduced vascular resistance and vasodilation, which directly affect DBP responses. [35]. This may provide a rationale for the higher DBP values observed in low-intensity exercise compared to high-intensity exercise. This finding is in accordance with de Souza et al. [36], who reported that hypertensive subjects exhibited higher blood pressure values (+2 mmHg) when exercising at 40% of their 1RM compared to 80%. Moreover, Orsano et al. [12] studied the acute blood pressure response to a session of high-velocity resistance training at a moderate intensity (70% of 10RM) and found a significant increase in SBP and DPB, corroborating the results of the current study.

Another noteworthy finding of the present study was that the deadlift and squat elicited the greatest increases in SBP, irrespective of training session intensity. This may be explained by the fact that these exercises recruit a larger amount of muscle mass compared with others. Moreover, they are generally classified as closed kinetic chain movements, in contrast to exercises such as the bench press and rowing, which are considered open kinetic chain movements. The squat primarily recruits the quadriceps, gluteus maximus, and gastrocnemius, while the deadlift emphasizes the hamstrings, gluteus maximus, and spinal erectors. Conversely, rowing predominantly activates the latissimus dorsi, rhomboids, trapezius, and biceps, while the bench press recruits the pectoralis major, deltoids, and triceps, thus involving comparatively smaller muscle groups. The greater muscle mass activated during squat and deadlift likely contributes to the amplified pressor response [37]. Furthermore, the mechanical characteristics of these exercises, particularly the involvement of hip flexion and extension, have been associated with increased cardiac output due to orthostatic stress [38]. Redistribution of blood flow toward active skeletal muscle should also be considered [39], as the recruitment of larger muscle groups is linked to greater cardiovascular stimulation and blood pressure responses [40]. These findings are consistent with Haennel et al. [41], who reported that cardiovascular stress during exercise is proportional to the active muscle mass, independent of contraction velocity, although their study focused on isokinetic exercises.

The present results have practical implications for managing patients with hypertension. The rise in blood pressure should be recognized as a typical and desirable physiological response, reflecting part of the adaptive process to resistance training. However, therapists should assess baseline blood pressure prior to each session, particularly when prescribing squat and deadlift exercises, ensuring that systolic values do not exceed 220 mmHg in men and 200 mmHg in women, as recommended by Niebauer et al. [42]. Importantly, despite the observed increases in SBP, both systolic and diastolic values remained within safe limits in our participants.

Medication use must also be considered, as pharmacological agents can modulate hemodynamic responses to exercise. The results of the current study showed that ARB use was associated with larger reductions in diastolic blood pressure after exercise, whereas ACE inhibitors, diuretics, and calcium channel blockers were associated with larger increases in blood pressure compared with participants using other medication classes. These findings underscore the importance of individualized monitoring, as patients may exhibit variable hypertensive or hypotensive responses depending on their medication profile.

### Limitation

Despite the results, some limitations must be pointed out. In the current study, it was not possible to measure blood pressure continuously during movement execution due to technical limitations of the device used. This measurement would have allowed us to analyze the blood pressure response to different phases of muscle contraction and to determine the peak for each exercise, which also could be useful for patients and therapists to identify the important moments of rise and fall in blood pressure. Therefore, future studies should consider continuously monitoring the non-invasive blood pressure during resistance training sessions. Another limitation that could be mentioned is that the sample size is small, which prevents us from extrapolating the results to all populations with hypertension and from performing subgroup analysis considering sex or age categories. Despite the limitation, by performing a crossover RCT, we ensure that the sample is homogenous for the comparisons between intensities and exercise types, which were the main focus of the study.

## 5. Conclusions

In conclusion, squats and deadlifts performed using a velocity-based protocol elicited the highest systolic and diastolic blood pressure responses, regardless of intensity. Additionally, diastolic blood pressure was lower following high-intensity exercises compared to the low-intensity protocols. Future studies should investigate the chronic effects of velocity-based resistance exercises on the blood pressure of individuals with hypertension.

## Figures and Tables

**Figure 1 jfmk-10-00349-f001:**
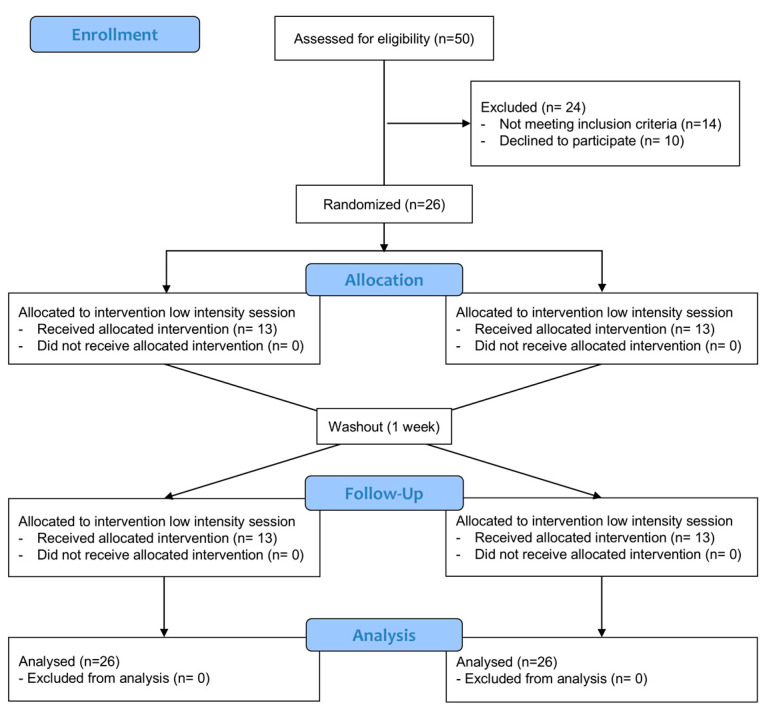
CONSORT flow diagram.

**Figure 2 jfmk-10-00349-f002:**
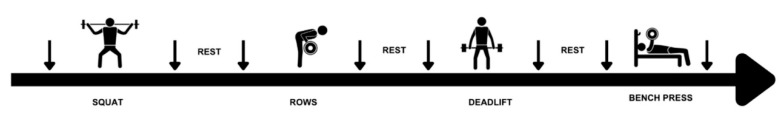
Intervention outcomes for all cognitive and psychosocial variables stratified by group.

**Table 2 jfmk-10-00349-t002:** Blood pressure assessed before and immediately after the exercise resistance training.

		High Intensity	Low Intensity	Intensity	Exercise Type	Interaction
				*p*-Value		
SBP (mmHg)	Baseline	121 ± 12	124 ± 10	0.60	<0.0001	0.25
	Squat *^,α^	134 ± 18	138 ± 15			
	Rows *^,†,α^	132 ± 14	132 ± 12			
	Deadlift *	138 ± 15	137 ± 11			
	Bench Press ^†^	126 ± 15	127 ± 10			
DBP (mmHg)	Baseline	81 ± 8	83 ± 9	0.003	<0.0001	0.12
	Squat ^α,#^	83 ± 9	86 ± 9			
	Rows	77 ± 10	83 ± 8			
	Deadlift ^α,#^	83 ± 8	86 ± 7			
	Bench Press	78 ± 8	79 ± 6			

SBP: systolic blood pressure; DBP: diastolic blood pressure; * *p* < 0.05 vs. Baseline; ^†^ *p* < 0.05 vs. Deadlift; ^α^ *p* < 0.05 vs. Bench press; ^#^ *p* < 0.05 vs. Rows.

**Table 3 jfmk-10-00349-t003:** Delta change in systolic and diastolic blood pressure calculated as values obtained after each exercise minus values obtained at baseline.

	** **	**SBP (mmHg)**		**DBP (mmHg)**	
	**Exercise**	**Delta Change (Δ)**	**Cohen’s d**	**Delta Change (Δ)**	**Cohen’s d**
High Intensity	Squat	13.2 ± 13.9	0.86	2.5 ± 6.8	0.24
	Rows	11.6 ± 10.4	0.84	−3.5 ± 6.5	−0.46
	Deadlift	17.7 ± 10.7	1.20	2.3 ± 6.2	0.25
	Bench Press	30.0 ± 5.6	0.30	−2.3 ± 6.2	−0.39
Low Intensity	Squat	14.8 ± 11.2	1.10	3.3 ± 7.8	0.34
	Rows	8.5 ± 8.5	0.70	−0.3 ± 7.4	0.05
	Deadlift	13.7 ± 11.6	1.20	3.2 ± 9.7	0.38
	Bench Press	2.9 ± 10	0.30	−4.0 ± 6.9	−0.50

SBP: systolic blood pressure; DBP: diastolic blood pressure.

**Table 4 jfmk-10-00349-t004:** Simple linear regression between blood pressure delta (**Δ**) change and medication use.

**Condition**	**Exercise**	**Dependent Variables**	**Independent Variables** **Medication Use**	**r^2^**	**B**	***p*-Values**
Low Intensity	Deadlift	ΔSBP	Diuretic	0.17	17.63	0.04
		ΔDBP	ARB	0.23	−9.78	0.01
		ΔDBP	ACE inhibitors	0.17	10.91	0.04
High Intensity	Bench Press	ΔDBP	ARB	0.19	−6.75	0.02
		ΔDBP	ACE inhibitors	0.16	7.48	0.05
		ΔDBP	Diuretic	0.17	9.54	0.04
		ΔDBP	CCB	0.17	9.54	0.04

SBP: systolic blood pressure; DBP: diastolic blood pressure; ARB: angiotensin-receptor blocker; ACE: angiotensin-converting enzyme; CCB: calcium channel blockers.

## Data Availability

The data supporting the findings of this study are available from the corresponding author upon request.

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
