# Peer review of "Acute Blood Pressure Changes Following Resistance Exercise in Adults with Hypertension"

_jfmk, 2025, doi:10.3390/jfmk10030349_

Round 1
Reviewer 1 Report
Comments and Suggestions for Authors
The study is of great interest because. The related topics covered
Summary
The echo given to the finding with such a small sample is striking
Introduction
It lacks more argumentation, it is only limited to the type of exercise but not to the profile of the subjects. The responses produced are heterogeneous both in terms of profile and program. The internal load per subject is different. Making a distinction by sex from the literature consulted would be appropriate.
They should provide what is the main hypothesis
Methods
With a standard deviation of 9 years and mean of 47 there cannot be subjects of 30, review it.
The anamnesis and anthropometry have not been explained, give some detail.
The reason and order of those proposed exercises should be explained (not the load which is) this detail is not justified.
Discussion
The quality of strength is measured by speed, so comparatively pressure changes with lower load and higher speed are not uncommon. There is no age distinction (the range is 30 years according to the methodology). Although the subjects are hypertensive their experience in the practice of physical activity is not indicated and is relevant to the study.
Given the small sample size, we suggest including it as an exploratory study and describing the findings as an interesting line to follow, avoiding making generalizations that cannot be made with such a small sample.
Author Response
|
Comments 1: The echo given to the finding with such a small sample is striking |
|
Response 1: We appreciate this suggestion. We reformulated the text avoiding generalization and considered the sample size as a study limitation (see abstract and study limitation page:10; Line 20) |
|
Comments 2: It lacks more argumentation, it is only limited to the type of exercise but not to the profile of the subjects. The responses produced are heterogeneous both in terms of profile and program. The internal load per subject is different. Making a distinction by sex from the literature consulted would be appropriate. |
|
Response 2: We thank the reviewer for this comment. As suggested, we added a paragraph in the introduction addressing this topic (page:2; Line:15). Moreover, this led us to perform some linear regression analysis to assess if some of these cofounding factors may explain the blood pressure response observed in the study. Since this required us to perform 96 linear regression analysis, we are now providing a supplementary file to show the results, and only the significant results were addressed in the manuscript’s text. Comments 3: They should provide what is the main hypothesis Response 3: Thank you for this comment. We added the hypothesis in the introduction section (Introduction last paragraph) (page:2; Line:20). Comments 4: With a standard deviation of 9 years and mean of 47 there cannot be subjects of 30, review it. Response 4: Thank you for this observation. However, standard deviation is not the same as minimum and maximum. Standard deviation of 9 years means that 68% of the participants will have an age between 38 and 56 years old. Therefore, it is totally plausible to have subjects with 30 years old and a mean of 47 and standard deviation of 9. Even though, we doubled check and the values are correct. Comments 5: The anamnesis and anthropometry have not been explained, give some detail. Response 5: Thank you for this comment. We provided more details about anamnesis and anthropometry (Page:3 Lines:24) Comments 6: The reason and order of those proposed exercises should be explained (not the load which is) this detail is not justified. Response 6: Thank you for this comment. The exercises were randomly assigned, following an alternating sequence of lower- and upper-body movements. This sequence was maintained across all subsequent sessions. This information was added to the Methods Section (Page:4; Line:28) Comments 7: The quality of strength is measured by speed, so comparatively pressure changes with lower load and higher speed are not uncommon. Response 7: Thank you for this comment. We cited in the discussion the study by Orsano et al., (2018) who found that high velocity training elicited increased blood pressure, being in accordance with our results (page:9 lines:27 ). Comments 8: There is no age distinction (the range is 30 years according to the methodology). Response 8: As replied above, we performed a correlation analysis between age and blood pressure delta changes for both intensities and all the exercise types. No significant association was found, suggesting that age did not play a role in our results. This was addressed in the results section. Comments 9: Although the subjects are hypertensive their experience in the practice of physical activity is not indicated and is relevant to the study. Response 9: Thank you for this comment. As inclusion criteria, participants must not have been engaged in regular physical activity for at least three months prior to the start of the study”. This information was added to the methods section (Page: 3 Line:4) Comments 10: Given the small sample size, we suggest including it as an exploratory study and describing the findings as an interesting line to follow, avoiding making generalizations that cannot be made with such a small sample. Response 10: We agree with the reviewer. We addressed this point as a study limitation, and the discussion was reformulated to avoid generalizations. |
Reviewer 2 Report
Comments and Suggestions for Authors
- Although statistically significant changes are reported, the manuscript does not address whether these changes are clinically meaningful in patients with controlled hypertension. Please include effect size measures.
- Explain how the number of participants was determined and whether a priori power analysis was performed.
- There are inconsistencies and typographical errors: “whit” instead of “with”
- Clarify whether the type of antihypertensive medication was considered as a covariate in the analysis, given its potential influence on hemodynamic responses.
- Discuss potential confounders such as time of day, nutritional status, habitual physical activity level, and adherence to pharmacological treatment.
- The results table presents data with unclear notations and subscripts. Please adhere to the journal’s formatting guidelines for improved readability.
- Link the results to specific recommendations from the American College of Sports Medicine (ACSM) and the European Society of Hypertension (ESH) regarding resistance training in hypertensive populations.
- Some references appear duplicated: references 17 and 19 and should be verified for both formatting and content compliance with journal requirements.
- Include concrete recommendations for health professionals and trainers on the safe prescription of resistance exercises in this population, particularly considering exercises that elicit greater pressor responses
Author Response
Comments 1: Although statistically significant changes are reported, the manuscript does not address whether these changes are clinically meaningful in patients with controlled hypertension. Please include effect size measures.
Response 1: Thank you for this comment. We added a new table (Table 3) presenting the delta change values for each exercise regarding the baseline, along with their corresponding effect sizes.
Comments 2: Explain how the number of participants was determined and whether a priori power analysis was performed.
Response 2: We appreciate this comment. Indeed a priori sample size calculation was performed. We provided more details on the sample size calculation (Page: 2 Lines:38)
Comments 3: There are inconsistencies and typographical errors: “whit” instead of “with”
Response 3: Thank you for this comment. This was corrected, and the manuscript was fully revised.
Comments 4: Clarify whether the type of antihypertensive medication was considered as a covariate in the analysis, given its potential influence on hemodynamic responses
Response 4: In order to account for possible confounding factors in the blood pressure response to exercise, we performed simple linear regression analyses considering delta SBP and delta DBP for each exercise (dependent variables) and the independent variables BMI, age, Sex (M/F), ARB use (Y/N), ACE inhib use (Y/N), Calcium blocker channel (Y/N) and diuretics (Y/N). Again, no significant associations were found, suggesting that these factors were not associated with the blood pressure response to exercise. Since this implies 96 linear regression analysis, and it serve just as a way to control for possible confounding factor, we presented a table with these results as supplementary file.
Comments 5: Discuss potential confounders such as time of day, nutritional status, habitual physical activity level, and adherence to pharmacological treatment.
Response 5: Thank you for raising this point. regression analyses were conducted to check for potential influence of BMI, Medication, Age and Sex on the SBP and DBP delta changes after each exercise, but no significant association was found. Likewise, we added a supplementary document with the new data. Regarding the physical activity level, as an inclusion criterion, participants should not have been engaged in regular physical activity at least 3 months prior the study. Concerning the adherence to pharmacological treatment, this was asked in during the anamnesis and also, as an inclusion criteria, participants should not have change their medication in the last 3 months.
Comments 6: The results table presents data with unclear notations and subscripts. Please adhere to the journal’s formatting guidelines for improved readability.
Response 6: Thank you. The table was corrected.
Comments 7: Link the results to specific recommendations from the American College of Sports Medicine (ACSM) and the European Society of Hypertension (ESH) regarding resistance training in hypertensive populations.
Response 7: Thank you for observation, the recommendations of the American College of Sports Medicine are addressed in the discussion (page:8; Line:13).
Comments 8: Some references appear duplicated: references 17 and 19 and should be verified for both formatting and content compliance with journal requirements.
Response 8: The changes were made.
Comments 9: Include concrete recommendations for health professionals and trainers on the safe prescription of resistance exercises in this population, particularly considering exercises that elicit greater pressor responses
Response 9: Thank you for this comment, The discussion proposes a clinical approach to hypertension that may support healthcare professionals in the effective management of this condition (page:9; Line:43).
Round 2
Reviewer 1 Report
Comments and Suggestions for Authors
The authors have made the requested modifications. I have no further comments
Author Response
Thank you for your comments, which have been valuable in improving and strengthening the article.
Reviewer 2 Report
Comments and Suggestions for Authors
Dear Reviewers,
We would like to sincerely thank you for the time and effort dedicated to reviewing our manuscript entitled: Acute Blood Pressure Changes Following Resistance Exercise in Adults with Hypertension.
We carefully considered all of your valuable comments and suggestions. In the revised version of the manuscript, we have addressed the points raised in detail, improving the methodological clarity, the quality of figures and tables, and the discussion of our findings in relation to the existing literature. Additionally, we have revised the language and style throughout the manuscript, which has significantly enhanced its readability and scientific rigor.
We believe that these revisions have strengthened the quality of the paper, and we are grateful for your constructive feedback, which has been essential in refining our work.
Thank you again for your thoughtful review and support.
Sincerely,
The Authors
Author Response

(The authors gave the same response as above.)
